# Effect of Anionic and Zeolite Supplements and Oral Calcium Bolus in Prepartum Diets on Feed Intake, Milk Yield and Milk Compositions, Plasma Ca Concentration, Blood Metabolites and the Prevalence of Some Reproductive Disorders in Fresh Dairy Cows

**DOI:** 10.3390/ani12213059

**Published:** 2022-11-07

**Authors:** Mohammad Mahdi Masoumi Pour, Farhad Foroudi, Naser Karimi, Mohammad Reza Abedini, Kazem Karimi

**Affiliations:** Department of Animal Science, Varamin-Pishva Branch, Islamic Azad University, Varamin 33817-74895, Iran

**Keywords:** anionic and zeolite, blood metabolites, dairy cows, hypocalcaemia, milk traits, reproductive disorders

## Abstract

**Simple Summary:**

We fed dairy cows diets with anionic salts and synthetic zeolite with an oral Ca bolus before calving and compared their feed intake, milk yield and compositions, plasma Ca concentration, blood metabolites and the prevalence of some reproductive disorders. There were significant differences in the milk production traits among groups, and the highest values belonged to the anionic group. The plasma Ca concentrations of the zeolite and anionic groups were higher than the control group during calving time and were for hours after that. The frequency of hypocalcaemia and reproductive disorders was also lower in the mentioned groups. The levels of blood metabolites in the experimental groups were also significantly lower than in the control group, and the use of an oral Ca bolus caused a further decrease in the values. Therefore, supplementation of anionic salts and synthetic zeolite, along with the use of an oral Ca bolus, has positive effects on milk production traits and improves the health and reproduction traits of fresh dairy cows.

**Abstract:**

Hypocalcaemia is an important disorder associated with an increased risk of metabolic diseases, and many studies have been going on for decades. This study investigated the effects of anionic and zeolite with an oral Ca bolus in the prepartum phase on milk yields and compositions, and plasma Ca concentrations, blood metabolites and the prevalence of some reproductive disorders in dairy cows after parturition. Ninety pregnant non-lactating Holstein-Friesian cows were randomly assigned to three isoenergetic diets and their counterparts using an oral Ca bolus: (1) Control (CON): low Ca (0.44%); (2) Anionic (ANI): high Ca (1.00%); (3) Zeolite (ZEO): low Ca (0.44%). The feed and energy intakes of the CON and ZEO groups were higher than the ANI group. The raw milk yield, Pr%, and feed efficiency did not differ between the groups, while the corrected milk yield, fat% and total corrected milk yield at 305 days differed between groups. Total plasma and the ionized Ca concentration of the ANI and ZEO groups, at calving time and in 6, 12, 24 and 48 h after that, were higher than in the CON group. The frequencies of hypocalcaemia and reproductive disorders in the ANI and ZEO groups were lower than in the CON. The blood metabolite levels in the CON group were higher than in other groups. In conclusion, the ANI and ZEO diets and their counterparts supplemented with an oral Ca bolus improved the milk production traits, plasma calcium and metabolites concentration and also effectively prevented reproductive disorders.

## 1. Introduction

The pre-parturition phase, as a main part of the transition period, plays an important role in the health and sustainability of dairy production. The cows which have a suitable transition period and are not exposed to a variety of stresses; they will not suffer from related diseases or a reduction in their milk production [1]. During the transition period, most fresh cows will face a variety of problems involving hypocalcaemia, metabolic diseases and reproductive disorders [2,3].

Supplementing dairy cows with anionic salts during a close-up period (three weeks before calving) improves the Ca homeostasis, decreases the risk of reproductive diseases and enhances the cow’s health status in the early postpartum period [4,5]. Hypocalcaemia is a damaging metabolic disorder that is associated with increased cow-body demand for calcium in order to produce more colostrum and milk. A failure to respond to this request increases the risk of metabolic diseases, such as ketosis, fatty liver, retained placenta, displaced abomasum, metritis and mastitis, as well as decreased reproductive performance and decreased milk production [2,6]. Subclinical hypocalcaemia, or a low circulating Ca at 48 h post-calving, is a common metabolic disorder affecting 47% of mature dairy cows and 25% of heifers at or after the first parturition [7]. Two main strategies, including reducing the dietary cation–anion difference (DCAD, −100 to −150 mEq/kg) via pre-parturient feeding of anionic compounds and using low-calcium diets (Ca, 0.24% to 0.44% of dry matter intake (DMI)) a few weeks before parturition to overcome problems, have been proposed [5,7]. However, the disadvantages of these methods, over time, have limited their use because employment of low-calcium diets is difficult in practice, and the level of calcium intake should be lowered through reducing the feed intake or the use of low-quality feed ingredients [8]. A reduced DMI during the transition phase and at the beginning of the new lactation period is strongly associated with a decreased quantity and quality of colostrum production as well as reduced fetal growth [8]. The adverse effect of an anionic diet with a negative dietary cation–anion difference on prepartum DMI is also well documented [9]. Anionic salts should be consumed in a 3 weeks period before calving, so it will be difficult to consume them and maintain the proper levels in the diet. Unpalatable anionic salts cause a reduced DMI with a negative energy balance and create undesirable results [10,11].

There are alternative methods to increase the plasma Ca concentration: a direct Ca injection or using an oral Ca bolus at different times after parturition [12,13]. Scarce reports are available on the calcium dynamics. The blood Ca concentration is influenced by injection or bolus supply after calving in dairy cows [7]. Both Holstein and Jersey breeds and their increasing parity are known risk factors for hypocalcaemia [14]. Increasing the Ca concentration in the rumen and the gastrointestinal tract using Ca supplementation is important to improve Ca absorption [11,15]. However, the effects of dietary Ca supplements in cows fed anionic diets, pre-calving, and the effect of postpartum Ca administration on the plasma Ca concentration and Ca homeostasis are not well defined [11,16]. The short- and long-term effects of the different sources of Ca supplements on cows fed anionic diets during the prepartum period were evaluated by Jahani-Moghadam et al. [7], who reported that the plasma concentration of Ca was affected by a direct Ca injection or application of an oral Ca bolus at 48 h post-calving (*p* < 0.01). The mean Ca at 6 h was greater in the Ca injection group compared to the control group (2.34 and 2.01 mmol/L respectively; *p* < 0.002). The cows of control group had the lowest Ca concentration at 12 h after parturition compared with the bolus supply and Ca injection cows (1.90, 2.16 and 2.14 mmol/L, respectively; *p* < 0.02); a similar trend was observed 24 h post-calving (*p* < 0.02). Based on the findings of the report, Ca supplements to fresh cows as an oral bolus are recommended in comparison with the subcutaneous Ca injection.

Vagnoni et al. [14] conducted a study on the effects of postpartum supplemental oral Ca for dairy cows fed prepartum dietary acidogenic salts. They found that the approach to providing acidogenic salts may be effective. They reported preliminary evidence that shortening the time of application of this approach may enhance its beneficial effects. The responses to the prepartum acidogenic salts and postpartum oral calcium supplementation were different for the cows in the second parity compared to cows of third parity or more. Furthermore, there was an increased risk of low blood calcium, particularly in the older Jersey cows, when the prepartum urinary calcium excretion was low, or the colostrum production was high. Wilms et al. [17] investigated the blood calcium dynamics in cows receiving an aqueous calcium suspension as voluntary consumption or as a calcium bolus following parturition. They reported that treatment-by-time interactions were present for the blood Ca, glucose and urea concentrations. The blood Ca was relatively stable in the Ca-drink cows, while higher fluctuations were observed in the Ca-bolus cows. At 24 h after Ca administration, the blood Ca was greater in cows receiving the Ca drink than in cows receiving the Ca bolus.

The different dosages of subcutaneous Ca borogluconate, immediately post-calving on dairy cow health and productivity, were evaluated by Amanlou et al. [12]. The short-term effects of the oral Ca supplements and subcutaneous Ca borogluconate injection have been evaluated by Domino et al. [18] under the same management conditions.

As an innovative solution, a high-performance natural calcium binder is the ideal method for enhancing the plasma Ca concentration after parturition [19,20,21]. As a natural mineral binder, zeolites, usually clinoptilolites (a group of hydrous sodium aluminosilicate minerals), is well able to bind divalent cations and absorb fungal toxins or harmful gases as well as trap the excess products in their structure [22,23]. The binding of dietary and gastrointestinal calcium resulted in a reduced plasma calcium concentration and ultimately stimulated the release of bone calcium reserves at calving time [22,24]. The unique properties of clinoptilolites include the ability to lose and gain water reversibly, ion-exchange ability or as feed additives are well listed [25]. Moreover, further investigations showed that clinoptilolites could be used as a preventer of metabolic diseases; however, few findings have been reported on the efficacy of clinoptilolites during the transition period or late gestation of cows [22,23,26].

A report by Katsoulos et al. [26] indicated that the dietary administration of 2.5% clinoptilolite during a few weeks of the transition period is very effective in preventing milk fever and ketosis after calving. A further study proved that a daily intake of 200 g of clinoptilolite in the diet enhances the immune response of calves vaccinated against *E. coli*. Moreover, the rumen pH, when using a high-concentrate diet, was higher. The research findings indicated that oral supplementation with clinoptilolites would improve the colostrum quality of cows without adverse effects on their health. Increased absorption of immunoglobulins in the intestine as well as a reduced incidence of diarrhea, were observed when clinoptilolite was used in the diet [19,27]. 

During the transition period, synthetic zeolite-A was used to control the availability of dietary minerals (e.g., Ca, Mg and P) to investigate its effect on immune functions. The lower gene expression of the neutrophil inflammatory mediators due to the altered availability of dietary minerals was observed, which indicates that zeolite-A may control inflammation during the transition period [28]. 

A study of the methods that do not have such limitations seems very necessary. Therefore, the objective of the current experiment was to evaluate the plasma Ca concentrations following the use of an oral Ca bolus in two anionic and zeolite diets fed to cows in the prepartum phase and its effect on feed intake, milk production and compositions, as well as the plasma Ca concentration and the frequency of some reproduction disorders. We hypothesized that the simultaneous use of an oral Ca bolus and zeolite supply would benefit from the independent use of each of them, and this was also compared to anionic salts. Therefore, the cows would have greater plasma Ca during different times after parturition with a better productive and reproductive performance.

## 2. Materials and Methods

### 2.1. Cows, Housing and Diets

This experiment was carried out on a commercial dairy farm (Fashafoyeh agriculture and Industry Co., Pakdasht, Tehran, Iran) from June to August 2020, with a population of 2300 lactating cows. The cows’ mean body condition score (BCS) at the beginning of the experiment was 3.50 ± 0.25, and the average milk yield was 40 ± 2.20 kg/day. The Thermal Humidity Index, during the experiment, was between 71 to 83. After calving, cows were housed in individual calving stalls for 48 h. Afterwards, they were moved to a postpartum pen for 21 days, where they were fed a total mixed ration (TMR) twice a day at 6 h a.m. and 14 h p.m. The diet was formulated according to the National Research Council [29] recommendations. The pre and postpartum diet ingredients, as well as the diet chemical composition, are shown in Table 1 and Table 2. Twenty-four days before the expected calving date, the cows were maintained in the close-up group and fed a ration containing anionic salts and zeolite supply with a dietary cation–anion difference (DCAD) of −100 and +100 mEq/kg DM, respectively. Ninety non-lactating multiparous Holstein-Friesian cows (mean BW = 754 ± 40 kg; mean parity = 3.00 ± 0.5) were randomly assigned to six experimental treatments or isoenergetic diets (three isoenergetic diets and their counterparts using the oral Ca bolus) based on their previous 305-day milk yield, body weight, BCS and number of parity as: (1) a control group without the oral Ca bolus (CON), Ca 0.44%, DCAD +100 mEq/kg; (2) a control group with the oral Ca bolus (COB), Ca 0.44%, DCAD +100 mEq/kg; (3) anionic without the oral Ca bolus (ANI), Ca 1.00%, DCAD −100 mEq/kg; (4) anionic with the oral Ca bolus (ANB), Ca 1.00%, DCAD −100 mEq/kg; (5) zeolite without the oral Ca bolus (ZEO), Ca 0.44%, DCAD +100 mEq/kg; and (6) zeolite with the oral Ca bolus (ZEB), Ca 0.44%, DCAD +100 mEq/kg (all dry matter basis). The oral bolus used in this research was manufactured by the Tosca Company, Tehran, Iran, containing 45 g of Ca (CaCl2, Ca-propionate and Ca fumarate). All oral Ca supplements were fed immediately after calving and at 6, 12 and 24 h post-calving. The anionic salts and zeolite supplement were used at 2.80% and 1.50% of the DMI basis, respectively. The used zeolite supplement (type A) was made by Sanayeh Shimiayiee Co., an Iranian chemical company (nAl_2_O_3_.mSiO_2_.xH_2_O; two-layer and eight-sided-type clinoptilolite with a zeolite/Ca binding ratio of 6:2 under the approved code: 2004-11-03 IZA).

### 2.2. Blood Analyses

Blood samples were taken into evacuated tubes containing heparin from the coccygeal vessels immediately after calving, after 6, 12, 24 and 48 h of parturition and 3 h after feeding. The samples were subsequently centrifuged at 3000× *g* for 15 min. The total plasma calcium concentration was measured by the photometric method (kits were from the Pars Azmoon chemical company, Tehran, Iran, TS.M.91.13.4) according to the manufacturer’s instructions using an analyzer (Roche Hitachi-911, Chemistry Analyzer Company). The plasma ionized calcium concentration was measured using the ion-selective electrode (ISE) method with an electrolyte analyzer (AC-9800, Audicom Company). The concentrations of β-hydroxybutyric acid (BHBA) and non-esterified fatty acid (NEFA) were determined by an enzymatic colorimetric method using commercial kits (BHBA: Abbott Diabetes Care Ltd., Witney, UK; NEFA: Cat. No. FA 115, Randox Laboratories Ltd., Crumlin, UK) according to the manufacturer’s instructions and the same auto-analyzer. All analyses were performed in duplicates. Before any samples were processed, the analyzer was calibrated with control sera containing known concentrations of N and P (TrueLabN^®^ and TrueLabP^®^, Pars Azmoon Co., Tehran, Iran) and a standard calibration solution (TrueCal U^®^, Pars Azmoon Co., Tehran, Iran).

### 2.3. Feed and Milk Analyses

Prior to the analyses, feed samples were ground through a 1 mm screen using a Wiley mill (Arthur Thomas Co., Philadelphia, PA, USA). The dry matter was determined by oven-drying at 65 °C until a constant weight was reached; method 930.15. For the crude protein (CP) analysis (N × 6.25), samples were analyzed by Kjeldahl titration (Kjeltec 1030 Auto Analyzer, Tecator, Höganäs, Sweden; [28]); method 920.53. The ether extract (EE) and ash contents were analyzed according to [30]; methods 920.39 and 941.12, respectively. The neutral detergent fiber (NDF) and acid detergent fiber (ADF) were analyzed according to [31] using heat-stable α-amylase (100 µL/0.5 g of sample) and sodium sulfite. All chemical analyses were performed in duplicates. The milk samples were tested for fat and protein percentages using an optical filter-based instrument (MilkoScan-605, Foss Electric, Hillerod, Denmark) [32].

### 2.4. Prevalence Reproductive Disorders

To calculate the prevalence of reproductive abnormalities in the cows of different experimental groups, the frequency (number of occurrences) of each case (abnormality) during the experiment was recorded and calculated as a corrected percentage in each group. Before data analysis, their normal distribution was tested, and discontinuous or numerical data were corrected using a standard statistical formula by SAS 9.2 software [31].

### 2.5. Statistical Analysis

All experimental data were analyzed by SAS 9.2 software [33] using a CRD statistical design. The data of continuous traits (e.g., feed and nutrient intake, milk production, Ca concentration and plasma metabolites) were analyzed by two-way variance analysis using the MIXED Procedure (Proc Mixed) with repeated observations. For discontinuous data or numerical traits (e.g., frequencies of reproduction disorders), nonparametric methods were employed. Before analyzing the discontinuous traits, data were corrected by the ArcSin√X formula. The Shapiro–Wilk test was also used to ensure the normal distribution of data. The mean differences in the treatments were compared at a level of *p* ≤ 0.01 using a Duncan’s multiple range test.

## 3. Results

### 3.1. Feed and Nutrients Intake, Milk Production Traits

Table 3 presents the results of the consumption of anionic salts and zeolite supplements with and without a bolus supply in different groups. During the prepartum phase, the DMI of the control group (CON) with 12.80 kg/day was the highest value (*p* ≤ 0.01) compared to the anionic (ANI) and zeolite (ZEO) groups with 12.44 and 12.36 kg/day, respectively. The energy intake of the groups also showed a similar trend with DMI; thus, the mean energy intake of the CON group was higher than the ANI and ZEO groups (*p* < 0.01). There were no significant differences between the experimental groups for the intake of CP, NDF, ADF and feed efficiency. On the 60th day of postpartum, the CON, COB, ZEO and ZEB (zeolite supplemented with Ca bolus) groups had a higher amount of DMI than the ANI and ANB (Anionic supplemented with Ca bolus) groups (*p* ≤ 0.01), with a difference of about 0.43 to 0.59 kg/day. Similarly, they had a higher amount of energy intake than the mentioned groups (*p* ≤ 0.05). A similar trend was observed in terms of the CP, NDF, ADF intake and values of FE, and there were no significant differences between them. Moreover, there was no significant difference between groups regarding the raw milk yield (RMY), but since the fat percentage of the groups was different, this difference was observed in the corrected milk yield (CMY). In this case, unlike the previous traits, the highest values of CMY (36.25 and 36.15 for ANI and ANB vs. 35.10, 35.11, 34.98 and 35.20 kg/day for the CON, COB, ZEO and ZEB groups, respectively) were for the ANI and ANB groups (*p* ≤ 0.01). Similarly, the fat percentages of the ANI and ANB groups were higher than other groups (*p* ≤ 0.01), but the protein percentage and feed efficiency had no significant difference between the groups. The total corrected milk yield was significantly different between groups (*p* ≤ 0.05), and the highest production values belonged to the anionic and anionic + bolus groups (11,056 and 11,026 kg/period), respectively. Therefore, the positive effects of the anionic and zeolite diets on the DM and nutrient intake, and also milk production traits of experimental cows, were confirmed.

### 3.2. Plasma Calcium Concentration

The effects of the experimental diets with and without the Ca bolus supply on the plasma calcium concentration of the cows at prepartum, parturition time and 48 h after that are presented in Table 4. During the prepartum phase, there was no significant difference between the groups in terms of the total plasma calcium concentration (Ca), ionized calcium concentration (Ca^++^) and the ratio between them (Ca^++^/Ca). At the parturition time, the total plasma Ca concentration of the ANI group was in the top (7.47 mg/dL) and for the CON and COB groups, they were at a minimum value (7.33 mg/dL) (*p* < 0.01). The other groups had no significant difference in this respect. A similar trend for Ca^++^ was observed (*p* < 0.01), but the ratio of Ca^++^/Ca between the groups was insignificant. At 6 h postpartum, the highest values of Ca, Ca^++^ and Ca^++^/Ca belonged to the zeolite groups and, then, to the anionic groups (*p* ≤ 0.01). The control groups, in this sense, were at minimum values compared to the other groups (*p* < 0.01). This trend was also observed with slight differences up to 48 h after parturition (*p* ≤ 0.01). By this means, the positive effects of the anionic and zeolite diets, supplemented with the Ca bolus, on the plasma calcium concentration of experimental cows were confirmed.

### 3.3. Prevalence of Hypocalcaemia and Reproductive Disorders

The effect of the experimental diets on the frequency of severe and subclinical hypocalcaemia, as well as postpartum reproductive disorders, are presented in Table 5. The corrected percentage of the incidence of severe hypocalcaemia for the CON, COB, ANI, ANB, ZEO and ZEB groups was 40.45, 38.56, 36.27, 33.21, 33.23 and 27.33%, respectively; therefore, the differences between the groups were significant (*p* < 0.01). A downward trend was observed among the CON to ZEB groups for this trait. The same trend, to some extent, existed for most of the reproduction traits (*p* ≤ 0.01). Regarding traits such as calving difficulty, abomasal displacement and culled cows, the anionic and zeolite groups did not show any cases compared to the control (*p* < 0.01). Moreover, in this case, positive results were obtained from supplementing the Ca bolus with anionic and zeolite groups.

### 3.4. Blood Metabolites

The response of the blood metabolites to the different experimental diets is shown in Table 6. The plasma levels of beta-hydroxybutyrate (BHBA) and non-esterified fatty acid (NEFA) showed no significant differences among the groups during the prepartum period. However, the level of plasma metabolites in the ANI and ZEO groups was lower than in the control groups. However, after parturition, the levels of the metabolites in the experimental groups were significantly lower than in the control group, and supplementing the diets with the Ca bolus caused a further reduction in the blood metabolite levels compared to the control group (*p* ≤ 0.01).

## 4. Discussion

### 4.1. Feed and Nutrients Intake, Milk Production Traits

In the prepartum, the DMI of the ANI and ZEO groups was significantly lower than the CON group. The lack of palatability of anionic salts is the main reason for a reduction in the DMI in the anionic groups and was predictable [5]. However, about reducing the DMI of the ZEO group, cannot be given a specific reason. However, in some reports, the accumulation of indigestible substances and their disturbing impact on the normal movements of the rumen is stated as the reason for reducing activity and feed intake. Correspondingly, it is reported that zeolite supplementation led to a significantly reduced ruminal dry matter digestibility and fermentation of organic matter [21,34]. Other parameters, such as energy, protein, NDF and ADF as the secondary components which follow the DMI, had a similar trend [7] but did not have significant differences in terms of the energy, protein, NDF and ADF intakes among groups. In the postpartum period, the DMI of the ANI group continued to decrease, but for the ZEO group, the DMI increased up to the CON group. In addition, the use of a bolus increased the dry matter and energy intake (*p* ≤ 0.05), as well as the protein, NDF and ADF intakes in the ZEO group. These results probably indicate the better efficiency of zeolite in promoting the feed intake and utilization of feed ingredients. Some studies have been reported on the efficacy of clinoptilolites on milk production traits in postpartum [20,26]. The most important functions of clinoptilolites are introduced as feed additives to improve reproductive traits. However, further investigations showed that clinoptilolites could be used for the prevention of metabolic diseases, which indirectly affect productive traits [22,23,26].

The DMI of the zeolite groups at postpartum period was increased up to the control group due to the impact of zeolite on ruminal activity, improve digestion and increase the passing rate of feed ingredients from the gastrointestinal tract [21,23]. There is no detailed information on how zeolite affects rumen activity and the fermentation process. However, due to the type of chemical structure and the buffering nature of zeolite, it has characteristics such as high water absorption power, creating an osmotic environment due to its high potential in absorbing positively charged ions (potassium, calcium, magnesium and ammonium), changing the type and population of microbes, and changing the production path of volatile fatty acids in the rumen which are the main effects attributed to the consumption of zeolite in various reports [10,22,23,24].

According to our results, the DMI of the cows increased after the consumption of zeolite (with or without bolus). This is probably due to the changes in the normal activity and passage rate through the digestive system, affecting feed consumption. In some reports, reducing feed intake and the passing rate has been mentioned due to the change in the conditions of the rumen environment, the balance of positively charged ions, water absorption and, to some extent, the accumulation of indigestible material in the rumen [20,22,23,24,25,26,27].

The research results indicate that when the pH of the rumen reaches less than 2.6, the survival of cellulolytic bacteria and fiber digestion decreases. These changes have a negative effect on the fermentation process and its products in the rumen, but the consumption of zeolite has the advantage of preventing a reduction in the rumen pH, controlling the digestive conditions and protecting the beneficial microorganisms of the rumen [24,35]. There are conflicting findings about the effects of zeolite on feed and nutrient intake. Some reports confirm the positive effects of zeolite on the consumption of dry matter, NDF and ADF [22,35]; however, some others do not confirm this due to the effect of factors, such as the feed intake level and duration of zeolite consumption, or the time required to adapt to the zeolite [25,26].

Anionic salts and clinoptilolites moderate the DCAD and maintain the desired pattern of rumen fermentation which, in turn, causes balanced fatty acids production (acetate and butyrate) and regulates calcium homeostasis, which will eventually lead to an increase in milk production [6,36]. The current results indicate the positive effects of anionic and zeolite supplementation on feed and nutrient intake and the improvement in milk production traits. It should be noted that due to the nature of normal diets with a low Ca content, they are more palatable than the anionic diets with a higher DMI; however, handling and managing these types of rations is very difficult [6,15,37].

The raw milk yield of the groups was similar at 60 days postpartum, but due to different milk fat content, their corrected milk yield showed a significant difference. The values were higher for the ANI and ANB groups than for the other two groups. This is expected due to the higher percentage of fat in these groups. Thus, the 305-day total corrected milk yield was also different between the groups. The protein percentage and feed efficiency (raw and corrected) did not show any difference between the groups. The percentage of milk protein is largely related to genetic and nutritional factors [11,38], but in this experiment, the level of dietary protein in all groups was the same. In terms of feed efficiency, since the changes of both components of milk yield and DMI were coordinated, the ratio between them did not show a change. However, the numerical values of the feed efficiencies in the anionic groups were higher than in the other groups. The feed efficiency based on the raw or corrected milk yield in the experimental groups was not significantly different, which shows that almost the same amount of feed and energy was consumed per unit of milk yield in these groups.

The results of this study showed that the positive effect of zeolite in binding calcium and improving calcium homeostasis in the body has finally prevented the complications of milk fever and improved the physiological conditions of the body to produce milk with the right fat percentage in the maximum production capacity of the cows. Some reports indicate that the consumption of zeolite, with a gradual release of the nitrogen, nutrients and minerals required by the rumen microbial population, may increase the production of microbial protein and consequently increase the percentage of protein, lactose and SNF in milk [22,23]; however, such effects were not observed in this experiment. This is probably related to the limited period of zeolite consumption (2 weeks vs. 4 weeks) by the cows during the close-up period [27,35].

### 4.2. Plasma Calcium Concentration

The results of Table 4 showed that the plasma concentration of the total and ionized Ca on the seventh day before parturition did not show any significant difference between the groups, but at 6, 12, 24 and 48 h after parturition, the plasma Ca concentration was higher in the diets with zeolite. All the experimental groups had a normal plasma Ca concentration immediately after parturition. However, the cows of the CON/COB groups had a lower plasma Ca concentration at 6, 12, 24 and 48 h post-parturition than the other groups, which was in agreement with Domino et al. [18], Jahani-Moghadam et al. [7] and Vagnoni et al. [14]. An increase in the plasma Ca concentration following the bolus administration was expected because, according to the report by Khorasani et al. [39], the rumen Ca transportation depends on the level of Ca consumption. In addition, ruminal Ca uptake increases several times when the ruminal Ca concentration increases to 11 mmol/L [11,14]. Based on the report by Dhiman and Sasidharan. [40], the average increase in the serum Ca concentration was 0.1 mmol/L after oral Ca supplementation (each dose contained 54.52 g Ca). Oetzel [41] stated that one day after parturition, the average increase in the serum Ca concentration was 0.18 mmol/L by using four doses of CaCl2 (contained 54 g Ca/dose). In the study by Jahani-Moghadam et al. [7], the serum Ca concentration was 0.185 mmol/L in cows fed the bolus in comparison with the control group within 48 h postpartum. Generally, it takes 24 to 48 h for the intestinal Ca absorption and bone reabsorption mechanisms to be activated (Hesters et al. [42]). Several studies [7,12,13,43] indicated that using anionic salts in prepartum might not completely resolve the problem of hypocalcaemia on day one after parturition, so the use of a Ca bolus might be necessary for dairy cows.

The critical threshold for the occurrence of clinical and subclinical symptoms of hypocalcaemia has not yet been precisely determined, despite extensive research, and fluctuates widely according to the results of different reports [44,45,46]. Based on the report of Oetzel [44], the dangerous level of concentration of plasma Ca at the time of parturition, which leads to the occurrence of clinical symptoms, is less than or equal to 7.3 mg/dL (equivalent to 1.825 mmol/L), and for subclinical hypocalcaemia, it is less than or equal to 8.5 mg/dL (equivalent to 1.2 mmol/L). In the present study, according to the available results, the dangerous Ca plasma concentration threshold at the time of parturition was determined to be 7.3 and 8.5 mg/dL for the clinical and subclinical manifestations, respectively.

According to our results, involvement with clinical hypocalcaemia continued until about 6 h after parturition, and a number of cows with related symptoms were observed in all groups; thus, the situation for all groups was almost similar. From parturition onwards, an increase in the plasma Ca concentration was observed, which resulted in a decreased frequency of hypocalcaemia in different groups. At the same time, the highest Ca plasma level belonged to the groups with zeolite and then to the anionic groups. In most cases, the plasma Ca levels for the administrated groups with the Ca bolus were clearly higher than their partners. This condition was observed up to 48 h after parturition. Some reports emphasize that if the dangerous threshold of less than 8.5 mg/dL of plasma Ca is considered, the usual measures for preventing and controlling subclinical hypocalcaemia are not very effective [5,44,47]. Among the groups of the current experiment, zeolite and anionic always had higher levels than this threshold. Therefore, the efficiency of using zeolite and anionic salt compared to normal conditions (CON) was well confirmed by the results of this experiment. The results of Table 4 and Table 5 show the fluctuations of the Ca plasma concentrations and the frequency of hypocalcaemia with different manifestations in the cows of the control group, which were more than the cows consuming zeolite or anionic salts.

### 4.3. Prevalence of Hypocalcaemia and Reproductive Disorders

Table 5 shows the effect of diets containing anionic salt and zeolite supplements on the prevalence of reproductive abnormalities compared to the control group. All abnormalities, including severe and subclinical hypocalcaemia, calving difficulty, retained placenta, endometritis and abomasal displacement, had a positive reaction with the use of anionic salt or a zeolite supplement to the diet, and their frequency was lower than the control group. In addition, the groups administrated with the bolus also had a lower frequency compared to their partners. Moreover, the rate of successful pregnancy 150th days after parturition in the groups containing anionic salt and zeolite supplements was significantly higher than in the control group. The effect of using the bolus was also positive, with an increasing impact on this attribute. The positive effect of anionic and zeolite supplements (with and without bolus) in the current research was in line with a number of studies that implied such effects on productive and reproductive traits [1,14,17] as well as improvement of pregnancy [22,27,35].

The mean hypocalcaemia prevalence in the report by Jahani-Moghadam et al. [7] was 33%. Amanlou et al. [12] reported that 58.6% of cows fed DCAD diets pre-calving had a Ca level of less than 2.125 mmol/L immediately after calving. Leno et al. [41], in a large field study, reported that 64% and 29% of cows in the third lactation period fed with anionic salts had hypocalcaemia symptoms at 0.3 and 1 DIM, respectively. These findings may explain that anionic diets have mitigated only some parts of hypocalcaemia complication, and there are other factors affecting cows’ responses to anionic diets, which require further investigation. According to the study by Jahani-Moghadam et al. [7], the lowest prevalence of hypocalcaemia through the first 48 h was generally observed in the bolus group. Therefore, a bolus could be considered a pioneer prophylactic treatment for all cows fed anionic diets.

According to the available reports, the acceptable percentage of retained placenta is 20% [41,44,48]. According to the results of Table 6, only the group partners administrated with the bolus were less than this criterion, and the main groups all showed more than this point. Therefore, the results of the current research confirm that the use of a bolus, especially in the groups containing the zeolite supplement and then anionic salt, has a significant effect in reducing the incidence of retained placenta in experimental cows. The available findings revealed that the incidence of retained placenta in cows consuming diets with negative DCAD and having an appropriate Ca concentration is lower compared to other diets [16,38]. In the present study, the lower percentage of retained placenta is probably related to the more suitable plasma Ca concentration in the zeolite- and anionic-consuming groups.

The average percentage of endometritis in the experimental groups was higher than 26%, and for their partners supplemented with the bolus, it was about 17%, which was significantly lower than in the experimental groups. The highest percentage was observed in the cows of the CON group, and the lowest rate was in the cows consuming anionic salt. Calving difficulty and abomasal displacement in the CON cows were less frequent than other abnormalities; instead, no cases of mentioned abnormalities were observed with anionic and zeolite consumption. Probably, this finding is related to the better control of plasma Ca concentrations using anionic and zeolite and their positive effects on the reproductive and digestive systems of the cows. The results of the available reports indicate that the level of Ca intake has a direct effect on preventing the abovementioned abnormalities [2,6,48].

### 4.4. Blood Metabolites

The results of the current study indicated that the levels of blood metabolites in the experimental groups were significantly lower than in the CON group. Therefore, according to the results of Table 3, we can interpret that the negative energy balance and increased energy needs in the CON group were higher than the ANI and ZEO groups due to increased feed or DMI than the other groups. Supplementation of the diets with Ca bolus causes no changes in DMI among groups. Meanwhile, the BHBA and NEFA levels of the ANI and ZEO groups were significantly lower than the CON group (Table 6), which confirms the result. Moreover, supplementing the groups with the Ca bolus did not cause a further reduction in the blood metabolite levels of the partners. Furthermore, the ANI and ZEO treatments influenced both BHBA and NEFA concentrations. There was also a significant effect of treatment × time interaction in a similar study [17], whereby the ANI and ZEO-treated cows (with or without Ca bolus) at 7 DIM had lower BHBA and NEFA levels than the CON group. Non-esterified fatty acid (NEFA) and beta-hydroxybutyric acid (BHBA) are the main blood metabolites that determine lipid mobilization in ruminants. Lipid is mobilized in the body to meet the energetic requirements of the cow during a period of negative energy balance in early lactation. Body fat is mobilized into the blood in the form of NEFA. The NEFAs are utilized to make about 40% of the milk fat during the first days of lactation. Plasma NEFA concentrations increase in response to increased energy needs due to inadequate feed and dry matter intake, and plasma NEFA concentrations usually are inversely related [49,50]. The negative energy balance in dairy cows induces lipolysis and lipid mobilization, resulting in more body fat decomposition and the downfall of the BCS. Evaluating the concentrations of NEFA and BHBA is good criteria for lipid mobilization and fatty acid oxidation in fresh dairy cows. NEFA reflects the magnitude of fat mobilization in body storage, and BHBA indicates the completeness of fat oxidization in the liver [6,50]. As the body fat storage becomes more decomposed, more NEFAs are taken up by the liver and the concentration of NEFA and BHBA in blood increases around parturition or in early lactation [49]. Negative energy balance and insufficiency of carbohydrate sources in the cow’s body around or after parturition leads to increased ketone body production, such as BHBA, which can result in ketosis. Furthermore, the increase in plasma NEFA concentration led to an increase in ketogenesis by hepatocytes [49,51,52].

According to the report by Jahani-Moghadam et al. [7], oral Ca supplementation did not result in any differences in the concentrations of glucose, NEFA or BHBA in serum compared with the un-supplemented cows. Similarly, Melendez et al. [17] and Dhiman and Sasidharan [40] reported no effect of oral Ca administration during the prepartum period on later concentrations of BHBA during the postpartum period. Lower blood NEFA and BHBA in Ca-supplemented cows have probably been driven by body mobilization, as determined by lower NEFA and BHBA at 7 DIM. In addition, an immediate effect of the supplemented diets by unknown mechanisms on milk production in the ANI and ZEO groups has probably led to lower BHBA and NEFA concentrations, which warrants more research in this area. Melendez et al. [17] found no effect of the Ca supplement on glucose, NEFA and BHBA when cows were treated by an oral Ca (CaCl2, Ca-propionate and propylene glycol) or by intravenous injection of Ca borogluconate. The lack of BHBA response to the Ca supplement is in agreement with previous experiments [12,45,46], though Goff et al. [52] showed a declined BHBA when an oral Ca supplement (as Ca propionate) was used. Nonetheless, increased postpartum DMI was not translated to any change in serum glucose or BHBA in other studies (Amanlou et al. [12]).

## 5. Conclusions

Compared to the control group, dietary Ca-limiting methods, such as the supplementation of zeolite and anionic salts with and without Ca bolus administration, improved the DMI and energy intake. The raw milk yield, protein percentage and feed efficiency were not different among the groups, while the corrected milk yield, fat percentage and total corrected milk yield at 305 days were higher for the anionic group. The plasma concentration of total Ca and ionized Ca after parturition was higher in the anionic and zeolite groups, and also, their frequencies for hypocalcaemia and reproductive disorders were lower than the control group. In addition, they had lower blood levels of BHBA and NEFA compared to the control group. Moreover, oral Ca bolus administration caused improved milk traits and decreased blood metabolite levels. In conclusion, the anionic and zeolite groups and their counterparts receiving the oral Ca bolus had positive effects on the milk yield and milk composition with better control gained for the plasma Ca concentration and metabolites levels as well as a preventive effect on the prevalence of reproductive disorders, and therefore, could be advised. However, due to the lack of information and insufficient experimental data in this field, more extensive research is needed.

## Figures and Tables

**Table 1 animals-12-03059-t001:** Feed ingredients of the experimental diets before and after parturition (% of DM unless otherwise noted).

Ingredients	Prepartum Diets			Postpartum Diet
	Control	Anionic	Zeolite	Normal
Alfalfa hay	16.11	15.81	15.85	17.20
Corn silage	31.11	30.52	30.64	28.70
Wheat straw	6.37	6.25	6.28	2.20
Wheat bran	3.94	0.22	3.88	-
Ground barley	11.82	11.81	11.65	13.50
Ground corn	11.96	11.94	11.78	15.80
Soybean meal 44%	10.94	12.93	10.77	11.93
Canola meal	3.57	2.68	3.52	3.95
Corn gluten meal 60%	1.88	1.78	1.85	1.95
Propylene glycol	0.98	0.93	0.97	1.10
Vitamin-mineral premix ^1^	0.98	0.93	0.97	1.20
Zeolite ^2^	-	-	1.50	-
Calcium carbonate	-	1.31	-	0.6
Magnesium oxide	0.34	0.33	0.34	0.3
Dicalcium Phosphate	-	-	-	0.22
Sodium Bicarbonate	-	-	-	1.20
Sodium Chloride	-	-	-	0.15
Calcium Chloride	-	0.85	-	-
Magnesium Sulphate	-	0.96	-	-
Calcium Sulphate	-	0.75	-	-
Total	100	100	100	100

^1^ Each kg of vitamin-mineral premix in experimental diets contained: 19 g Mg, 12 g Fe, 10 g Mn, 13 g ZN, 300 mg Cu, 100 mg Co, 30 mg Se, 100 mg I, 5 million IU vitamin A, 1 million IU vitamin D3 and 30 mg vitamin E. ^2^ Zeolite, manufactured by Iranian Chemical Industry (general formula: nAl_2_O_3_.mSiO_2_.xH_2_O), a two-layer and eight-sided class of clinoptilolite with zeolite/Ca binding ratio of 6:2. DM = 90%, DCAD = −3400 mEq/kg of DM.

**Table 2 animals-12-03059-t002:** Chemical composition of the experimental diets before and after parturition (% of DM unless otherwise noted) ^1^.

Nutrients	Prepartum Diets			Postpartum Diet
	Control	Anionic	Zeolite	Normal
Crude protein	16.24	16.24	16.24	16.54
NEL ^2^ (Mcal/kg)	1.58	1.58	1.58	1.61
NDF ^3^	35.50	35.30	35.50	31.74
ADF ^4^	21.00	21.00	21.00	19.22
NFC ^5^	38.70	38.00	38.70	40.57
EE ^6^	2.65	2.50	2.70	2.65
Ca	0.44	1.00	0.44	0.44
P	0.36	0.31	0.36	0.36
Mg	0.44	0.44	0.44	0.44
Cl	0.16	0.25	0.16	0.16
K	1.26	1.22	1.26	1.26
Na	0.05	0.05	0.05	0.05
S	0.22	0.28	0.21	0.22
DCAD ^7^ (mEq/kg DM)	+100	−100	+100	+300

^1^ Determined on samples pooled by week (n = 8). ^2^ NEL: net energy for lactation (based on tabular values; [27]), ^3^ NDF: neutral detergent fiber, ^4^ ADF: acid detergent fiber, ^5^ NFC: non-fiber carbohydrate (100 – NDF − neutral detergent insoluble CP) + (CP + ash + fat), ^6^ EE: ether extract, ^7^ DCAD: dietary cation–anion difference ((Na^+^ + K^+^) − (Cl^−^ + S^−2^)).

**Table 3 animals-12-03059-t003:** Comparison of feed and nutrient intake and milk production traits of experimental diets on day 60 after parturition (kg/day, unless otherwise noted).

Item	Diets							*p*-Value		
	CON	COB	ANI	ANB	ZEO	ZEB	SEM	Trt	Time	Trt × Time
Prepartum										
DMI (kg/day) ^1^	12.80 ^a^	-	12.44 ^b^	-	12.36 ^b^	-	0.163	<0.001	<0.011	<0.010
NE_L_ (Mcal/day) ^2^	20.22 ^a^	-	19.66 ^b^	-	19.51 ^b^	-	0.216	0.001	0.445	0.560
CP (kg/day) ^3^	2.08	-	2.02	-	2.00	-	0.236	0.518	0.368	0.685
NDF (kg/day) ^4^	4.29	-	4.13	-	4.38	-	0.223	<0.016	0.633	<0.018
ADF (kg/day) ^5^	2.69	-	2.52	-	2.59	-	0.158	<0.022	0.755	<0.025
Feed Efficiency	2.79	-	2.99	-	2.85	-	0.139	0.326	0.449	0.857
Postpartum	60 days postpartum									
DMI (kg/day) ^1^	23.41 ^a^	23.40 ^a^	22.86 ^b^	22.82 ^b^	23.29 ^a^	23.40 ^a^	0.151	<0.018	0.033	<0.014
NE_L_ (Mcal/day) ^2^	37.69 ^a^	37.67 ^a^	36.80 ^b^	36.74 ^b^	37.50 ^a^	37.67 ^a^	0.333	0.036	0.035	0.050
CP (kg/day) ^3^	3.87	3.87	3.78	3.77	3.85	3.87	0.155	0.118	0.168	0.385
NDF (kg/day) ^4^	7.43	7.73	7.26	7.24	7.39	7.43	0.203	0.511	0.345	0.660
ADF (kg/day) ^5^	4.50	4.50	4.39	4.39	4.48	4.50	0.152	0.418	0.268	0.785
Raw milk yield (kg/day)	42.37	42.36	42.72	42.68	42.19	42.31	1.22	0.001	0.001	<0.011
Corrected milk yield (kg/day) *	35.10 ^b^	35.11 ^b^	36.25 ^a^	36.15 ^a^	34.98 ^b^	35.20 ^b^	1.01	0.001	0.001	<0.012
Fat%	2.87 ^b^	2.86 ^b^	2.99 ^a^	2.98 ^a^	2.86 ^b^	2.88 ^b^	0.08	<0.021	0.000	<0.002
Protein%	2.26	2.27	2.29	2.30	2.36	2.35	0.11	0.322	0.562	0.652
Raw Feed Efficiency	1.81	1.81	1.87	1.87	1.81	1.81	0.14	0.377	0.501	0.555
Corrected Feed Efficiency	1.50	1.50	1.59	1.58	1.50	1.50	0.13	0.362	0.723	0.686
Total corrected milk yield at 305 days (kg) *	10,706 ^b^	10,709 ^b^	11,056 ^a^	11,026 ^a^	10,669 ^b^	10,736 ^b^	197	<0.020	0.023	<0.011

^ab^ The different superscript letters on the top right of the values in each row from left to right indicate a significant difference of means at *p* ≤ 0.01. Diets: CON: control, COB: control + bolus, ANI: anionic, ANB: anionic + bolus, ZEO: zeolite, ZEB: zeolite + bolus. SEM = Standard Error of Mean. * Milk yield based on standard fat 4% calculated from formula: FCM4% = 0.4 (kg Milk) + 15 (kg Milk × fat%) per day or per a period. Feed efficiency, calculated as: raw or corrected milk yield (kg/day)/DMI (kg/day). ^1^ DMI= Dry Matter Intake; ^2^ NE_L_ = Net Energy for Lactation; ^3^ CP = Crude Protein; ^4^ NDF = Neutral Detergent Fiber; ^5^ ADF = Acid Detergent Fiber.

**Table 4 animals-12-03059-t004:** Effect of experimental diets on plasma calcium concentration at parturition time and 48 h after that (mg/dL).

Item	Diets ^1^							*p*-Value		
	CON	COB	ANI	ANB	ZEO	ZEB	SEM ^2^	Trt	Time	Trt × Time
7 days prepartum										
Ca	9.30	-	9.27	-	9.38	-	0.131	0.533	0.478	0.655
Ca^++^	4.30	-	4.31	-	4.44	-	0.112	0.412	0.128	0.112
Ca^++^/Ca	0.46	-	0.46	-	0.47	-	0.016	0.415	0.354	0.215
At Parturition										
Ca	7.33 ^b^	7.33 ^b^	7.47 ^a^	7.43 ^ab^	7.41 ^ab^	7.42 ^ab^	0.100	0.000	<0.012	<0.022
Ca^++^	3.37 ^b^	3.36 ^b^	3.51 ^a^	3.52 ^a^	3.44 ^ab^	3.46 ^ab^	0.052	<0.018	<0.033	<0.025
Ca^++^/Ca	0.46	0.46	0.47	0.47	0.46	0.47	0.014	0.321	0.245	0.255
6 h postpartum										
Ca	7.68 ^c^	7.66 ^c^	7.89 ^b^	7.82 ^b^	7.95 ^a^	7.98 ^a^	0.136	0.000	<0.011	<0.011
Ca^++^	3.45 ^d^	3.47 ^d^	3.63 ^c^	3.73 ^b^	3.82 ^a^	3.68 ^bc^	0.017	0.000	<0.045	<0.044
Ca^++^/Ca	0.44 ^b^	0.45 ^b^	0.46 ^ab^	0.48 ^a^	0.48 ^a^	0.46 ^ab^	0.009	0.451	0.633	0.555
12 h postpartum										
Ca	8.36 ^c^	8.37 ^c^	8.38 ^c^	8.35 ^c^	8.53 ^b^	8.90 ^a^	0.134	0.000	<0.041	<0.035
Ca^++^	3.86 ^b^	3.88 ^b^	4.02 ^ab^	4.16 ^a^	4.10 ^a^	4.50	0.055	0.000	0.005	<0.017
Ca^++^/Ca	0.46 ^c^	0.46 ^c^	0.48 ^b^	0.50 ^a^	0.48 ^b^	0.51	0.007	0.000	0.285	0.911
24 h postpartum										
Ca	8.57 ^b^	8.63 ^b^	8.88 ^a^	9.06 ^a^	8.89 ^a^	9.01 ^a^	0.131	0.000	<0.050	<0.045
Ca^++^	4.02 ^c^	4.08 ^c^	4.45 ^ab^	4.45 ^ab^	4.27 ^b^	4.65 ^a^	0.055	0.000	<0.022	<0.037
Ca^++^/Ca	0.47 ^b^	0.47 ^b^	0.50 ^a^	0.49 ^ab^	0.48 ^ab^	0.51 ^a^	0.008	0.000	0.501	0.713
48 h postpartum										
Ca	8.77 ^b^	8.88 ^b^	8.87 ^b^	9.05 ^b^	9.12 ^ab^	9.26 ^a^	0.126	0.000	0.051	<0.044
Ca^++^	4.06 ^b^	4.16 ^b^	4.38 ^ab^	4.62 ^a^	4.38 ^ab^	4.41 ^ab^	0.050	0.000	0.002	<0.033
Ca^++^/Ca	0.46 ^b^	0.47 ^b^	0.49 ^ab^	0.51 ^a^	0.48 ^ab^	0.47 ^b^	0.008	0.000	0.551	0.555

^abc^ The different superscript letters on the top right of the values in each row from left to right indicate a significant difference of means at *p* ≤ 0.01. ^1^ Diets: CON: control, COB: control + bolus, ANI: anionic, ANB: anionic + bolus, ZEO: zeolite, ZEB: zeolite + bolus. ^2^ SEM = Standard Error of Mean.

**Table 5 animals-12-03059-t005:** Effect of experimental diets on prevalence of hypocalcaemia and reproduction disorders after parturition.

Frequency % ^1^	Diets ^2^							*p*-Value
	CON	COB	ANI	ANB	ZEO	ZEB	SEM ^3^	
Severe hypocalcaemia	40.45 ^a^	38.56 ^ab^	36.27 ^ab^	33.21 ^b^	33.23 ^b^	27.33 ^c^	2.330	0.000
Subclinical hypocalcaemia	34.19 ^a^	31.22 ^ab^	22.74 ^bc^	17.33 ^c^	26.56 ^b^	21.33 ^bc^	3.110	0.000
Calving difficulty	25.24 ^a^	25.24 ^a^	17.55 ^b^	0.00 ^c^	16.78 ^b^	0.00 ^c^	2.160	0.000
Retained placenta	31.10 ^a^	17.33 ^c^	26.56 ^b^	16.48 ^c^	26.56 ^b^	14.29 ^c^	2.250	0.000
Endometritis	33.21 ^a^	22.18 ^b^	26.56 ^ab^	14.52 ^b^	30.00 ^ab^	16.27 ^b^	2.540	<0.003
Abomasal displacement	17.55 ^a^	16.45 ^a^	0.00 ^b^	0.00 ^b^	0.00 ^b^	0.00 ^b^	1.750	0.000
Culled Cows	12.92 ^a^	0.00 ^b^	0.00 ^b^	0.00 ^b^	0.00 ^b^	0.00 ^b^	2.360	0.000
Pregnancy at 150 days	53.72 ^b^	55.78 ^b^	63.43 ^a^	65.91 ^a^	64.21 ^a^	65.91 ^a^	4.860	0.000

^abc^ The different superscript letters on the top right of the values in each row from left to right indicate a significant difference of means at *p* ≤ 0.01.^1^ Frequency noted based on the corrected percentage of animals which showed signs of abnormality. ^2^ Diets: CON: control, COB: control + bolus, ANI: anionic, ANB: anionic + bolus, ZEO: zeolite, ZEB: zeolite + bolus. ^3^ SEM = Standard Error of Mean.

**Table 6 animals-12-03059-t006:** Effect of experimental diets on blood metabolites of the cows.

Item	Diets							*p*-Value		
	CON	COB	ANI	ANB	ZEO	ZEB	SEM	Trt	Time	Trt × Time
Prepartum (7 days before parturition)										
BHBA	0.75	-	0.68	-	0.71	-	0.082	0.352	0.546	0.711
NEFA	0.29	-	0.26	-	0.28	-	0.022	0.511	0.233	0.211
Postpartum (7 days after parturition)										
BHBA	0.81 ^a^	0.78 ^ab^	0.75 ^b^	0.72 ^b^	0.75 ^b^	0.71 ^b^	0.042	0.000	<0.022	<0.037
NEFA	0.25 ^a^	0.22 ^ab^	0.18 ^b^	0.18 ^b^	0.17 ^b^	0.17 ^b^	0.011	0.000	0.000	0.013

^ab^ The different superscript letters on the top right of the values in each row from left to right indicate a significant difference of means at *p* ≤ 0.01. Diets: CON: control, COB: control + bolus, ANI: anionic, ANB: anionic + bolus, ZEO: zeolite, ZEB: zeolite + bolus. SEM = Standard Error of Mean.

## Data Availability

The data presented in this study are available on request from the corresponding author. The data are not publicly available due to privacy restrictions.

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
