# Peer review of "Effect of Anionic and Zeolite Supplements and Oral Calcium Bolus in Prepartum Diets on Feed Intake, Milk Yield and Milk Compositions, Plasma Ca Concentration, Blood Metabolites and the Prevalence of Some Reproductive Disorders in Fresh Dairy Cows"

_animals, 2022, doi:10.3390/ani12213059_

Round 1

Reviewer 1 Report

In this manuscript, Masoumipour et al. evaluated the effects of aniomic and zeolite supplementation and oral calcium bolus on fresh dairy cows. The experiments and data presented are clear. There are several points needed to be clarified. 

1.     How many cows were used in the experiments?

2.     The authors stated that they performed one-way ANOVA for statistical analysis. But in the table 3 and 4, how did they calculate Trt X Time? The interaction between treatment and time should be analyzed by two-way ANOVA.

3.     It is not clear what a,b,c mean in the table. Although they stated that the different superscript letters on the top right of the values indicate a significant difference of means at P≤0.01. Which groups does a or b or c compare?  Does a, b or c mean the p value is smaller than 0.01? What is the difference between a, b and c? What do ab and bc mean in table 4 and table 6? In the tales, what does P=0.000 mean?  

Grammar needs to be corrected. 

Author Response

First of all, the authors of the article should appreciate and thank the constructive and useful comments of the respected reviewer, and then the answer to the point of views are presented as follows.

Point 1: How many cows were used in the experiments?

Response 1:

The number of non-experimental cows of commercial dairy farm was 2300 heads. But the number of experimental cows was exactly 90, which were used in the 6 treatments with 15 observations per each treatment (1.control; 2. Anionic; 3. Zeolite without oral Ca bolus and their counterparts (3 treatments) with oral Ca bolus). We apologize for this mistake. The text error was corrected.

Point 2: The authors stated that they performed one-way ANOVA for statistical analysis. But in the table 3 and 4, how did they calculate Trt X Time? The interaction between treatment and time should be analyzed by two-way ANOVA.

Response 2:

Thank you. You are absolutely right. The interaction effects of two independent factors (treatment and time) should be analyzed in a two-way method, which was done, but it was wrongly stated in the statistical analysis section of the one-way method. The text error was corrected.

Point 3: It is not clear what a,b,c mean in the table. Although they stated that the different superscript letters on the top right of the values indicate a significant difference of means at P≤0.01. Which groups does a or b or c compare?  Does a, b or c mean the p value is smaller than 0.01? What is the difference between a, b and c? What do ab and bc mean in table 4 and table 6? In the tales, what does P=0.000 mean?  

Response 3:

  1. That phrase modified to "the different superscript letters on the top right of the values in each row from left to right indicate a significant difference of means at P≤0.01".
  2. Dfferent means with different superscript letters are compared based on the larger mean for "a" with larger value and smaller mean for "b and c …" with smaller values.
  3. In most cases a, b or c mean the p value is smaller than 0.01, and even lower probability levels, whose exact values are included in the mentioned tables for comparison.
  4. The only difference between the means is their numerical value, and this value is calculated and inserted according to the minimum significant difference which is reported by software.
  5. The expression ab or bc means the lack of significant difference of the corresponding mean with the numerical value of the mean greater or smaller than itself, which is reported by the software in the Duncan's mean comparison method (DMRT= Duncan's Multiple Range Test) of SAS.
  6. The P=0.000 mean?  It means the lowest error probability (P-value) of 0.000 (At the level of 1 to 10000, the next number is less than 5 and is not processed), based on the software output report.

Point 4: Grammar needs to be corrected. 

Response:

The text of the article was revised again and possible writing and grammatical errors were reviewed and corrected.

Reviewer 2 Report

in table 3 some parameters do not contain letters of statistical analysis

the authors must add updated references in 2022

Author Response

First of all, the authors of the article should appreciate and thank the constructive and useful comments of the respected reviewer, and then the answer to the point of views are presented as follows.

Point 1: in table 3 some parameters do not contain letters of statistical analysis

Response 1:

Thanks for the careful opinion of the honorable reviewer. The absence of a significant sign above some values ​​in the mentioned table means the absence of a statistically significant difference between the reported means. However, the data in Table 3 were completely controlled and revised.

Point 2: the authors must add updated references in 2022.

Response 2:

Thank you. An attempt was made to use newer sources in this regard. The result is clear in the reference section. But it should be noted that, there is not enough information in this field and it is one of the reasons and the necessity of research on this issue.

Reviewer 3 Report

The present study entitled “Effect of Anionic and Zeolite Supplemented Pre-Partum Diets and Oral Calcium Bolus on Feed Intake, Milk Yield and Compositions, Plasma Ca Concentration, Blood Metabolites and the Prevalence of Some Reproductive Disorders in Fresh Dairy Cows” is interesting, valuable, well written and within scope of journal. The study should be accepted for publication with minor concerns given below.

Abstract: authors are suggested to add one introductory line stating the problem or background of the study.

Line 72-74: rewrite this sentence, seems unclear

Line 136-137: what is average milk yield stand for? In a day or in a week?

Line 299: Is energy a nutrient?

Conclusion: authors should write limitation and future prospective of study.

Author Response

First of all, the authors of the article should appreciate and thank the constructive and useful comments of the respected reviewer, and then the answer to the point of views are presented as follows.

Point 1: Abstract: authors are suggested to add one introductory line stating the problem or background of the study.

Response 1:

Thanks for the useful suggestion of the respected reviewer. According to the recommendation, a line was added to the summary of the article regarding the statement of the problem and background.

Point 2: Line 72-74: rewrite this sentence, seems unclear.

Response 2:

The desired line was modified based on the opinion of the respected reviewer.

Point 3: Line 136-137: what is average milk yield stand for? In a day or in a week?

Response 3:

Apologies for the sentence error. In line 136-137 the average milk yield stated as average daily milk yield and was 40±2.20 kg/d. The text error was corrected.

Point 4: Line 299: Is energy a nutrient?

Response:

According to the definition of some scientific references, whose description is not possible in this short answer, the useful components in the food consumed by a living organism are called nutrients, which the organism needs for its survival and growth. "Macronutrients" are the main amount of water, energy, protein, etc. At the same time, the desired word and sentence was modified based on the opinion of the respected reviewer.

Point 5: Conclusion: authors should write limitation and future prospective of study.

Response:

It was done based on the opinion of the respected reviewer.